# Widely Targeted Metabolic Profiling Reveals Differences in Polyphenolic Metabolites during *Rosa xanthina* f. *spontanea* Fruit Development and Ripening

**DOI:** 10.3390/metabo12050438

**Published:** 2022-05-13

**Authors:** Yanlin Sun, Yu Yang, Meichun Zhou, Le Luo, Huitang Pan, Qixiang Zhang, Chao Yu

**Affiliations:** 1Beijing Key Laboratory of Ornamental Plants Germplasm Innovation & Molecular Breeding, Beijing 100083, China; syl980112@163.com (Y.S.); yangyu17710117767@163.com (Y.Y.); linliw2001@163.com (M.Z.); luolebjfu@163.com (L.L.); htpan@bjfu.edu.cn (H.P.); zqxbjfu@126.com (Q.Z.); 2National Engineering Research Center for Floriculture, Beijing 100083, China; 3Beijing Laboratory of Urban and Rural Ecological Environment, Beijing 100083, China; 4Engineering Research Center of Landscape Environment, Ministry of Education, Beijing 100083, China; 5Key Laboratory of Genetics and Breeding in Forest Trees and Ornamental Plants of Ministry of Education, Beijing 100083, China; 6School of Landscape Architecture, Beijing Forestry University, Beijing 100083, China

**Keywords:** *Rosa xanthina* f. *spontanea*, rose hips, ripening stages, polyphenols, metabolomics

## Abstract

Rose hips are rich in various nutrients and have long been used for food and medicinal purposes. Owing to the high phenolic content, rose hips can be used as natural antioxidants. In this study, ultra-performance liquid chromatography-tandem mass spectrometry (UPLC-MS/MS) was used to conduct a widely targeted metabolomics analysis on the polyphenolic components of *Rosa xanthina* f. *spontanea* in three ripening stages: unripe, half-ripe and fully ripe fruit. A total of 531 polyphenol metabolites were detected, including 220 phenolic acids, 219 flavonoids, 50 tannins and 42 lignans and coumarins. There were 160 differential metabolites between unripe and half-ripe rose hips (61 downregulated and 99 upregulated) and 157 differential metabolites between half-ripe and fully ripe rose hips (107 downregulated and 50 upregulated). The results of our study not only greatly enrich the chemical composition database of rose hips but also provide metabolomics information on the changes in polyphenolic metabolism during fruit development for the first time, which will help select the optimal harvest time of rose hips to achieve better quality.

## 1. Introduction

Rose hips, the pseudo fruit of *Rosa* genus in the Rosaceae family, have served as food and medicine for a long time [1]. They are utilized in a large variety of drinks and foodstuffs and to improve skin care, as well as to treat various ailments, including influenza, colds, inflammation, and chronic pain among others [2]. For example, rose hips can be consumed fresh or after processing, such as juice, dried and canned products, yogurt and wine, such as the fruit of *R. canina*, *R. rugosa*, *R. roxburghii* and *R. arvensis* [3,4,5]. In medicine, *R. laevigata* fruit is widely consumed in China for its medicinal effects, which include improving resistance to colds and kidney health, increasing sperm counts, reducing inflammation, and treating arteriosclerosis [6]. *R. canina* fruit have been used as a diuretic and laxative and to treat gout and rheumatism in traditional medicine [7]. 

The health benefits of rosehip can primarily be attributed to their high concentration of natural antioxidants, such as phenolic compounds, vitamin C and carotenoids [8]. Polyphenols are among the most abundant and widely distributed secondary metabolites in plants. Interest in them originates from their powerful antioxidant activity and various pharmacological properties that include anti-inflammatory, antiallergenic and antibacterial activities [9]. Rose hips are rich in some phenolic compounds, including catechin, myricetin, quercetin, procyanidin-B2, and ellagic, gallic, chlorogenic, caffeic, *p*-coumaric, ferulic and sinapic acids [2,10,11], which significantly contribute to human health. In recent years, owing to its high antioxidant capacity, increasing numbers of researchers have begun to study the polyphenolics in rose hips more closely. The content of polyphenolics will differ depending on the stage of ripening of the fruit. Understanding the physical and chemical characteristics and nutritional characteristics of fruit is critical for designing cost-effective and efficient post-harvest treatment equipment and optimizing biological processes and natural medicinal products in functional food manufacturing.

China is home to 95 species (65 endemic) of *Rosa*, accounting for almost half of the total in world. This country is one of the biodiversity centers of wild *Rosa* [12]. *R. xanthina* f. *spontanea* in the Rosaceae is widely distributed, rich in resources, high yielding and strongly resistant to cold and drought. The wild fruit is highly valuable with multi-functions of both medicine and foods, which has anti-aging, anti-oxidation, anti-fatigue and anti-thrombosis effects [13]. Currently, there are many studies on the components of rose hips, but most focus on the detection of several phenolic components, and the flux is low. To the best of our knowledge, no scientific research that encompass a systematic analysis of the metabolic characteristics and changes that occur during the *R. xanthina* f. *spontanea* ripening stages have been published.

Widely targeted metabolomics has been commonly applied to agriculture, food science and botany and can integrate the advantages of the non-target and targeted metabolic identification methods, achieve a high-throughput, and exhibit high sensitivity and wide coverage [14]. In this study, for the first time to the best of our knowledge, the polyphenolic components of *R. xanthina* f. *spontanea* fruit in different stages were detected by ultra-performance liquid chromatography-tandem mass spectrometry (UPLC-MS/MS). Furthermore, exploration of the polyphenol metabolites of different maturity stages of rosehips provides insights for proper use of each stage, which will guide experimentation and production.

## 2. Results

### 2.1. Measurement of Color

Fruit color changes were measured with a colorimeter, and the CIELab system was used to describe colors, which has also been applied to other fruit, such as strawberry, mango, and apple, to ascertain ripeness [15]. The rose hips were collected at three ripening stages: (1) unripe (green and hard, G), (2) half-ripe (red and hard, R), and (3) fully ripe (purple and soft, P). The fruit color parameters in each stage are shown in Table 1, with significant changes in external color between the different ripening stages (Figure 1). The decreasing *L** values illustrate that rose hips darkened during ripening, and the rose hips became less yellow as demonstrated by a slight decrease in the *b** values. This corresponds to the change in color parameters of *R. canina* when it matures [16]. The *a** values increased dramatically from the unripe to half-ripe stage and then decreased significantly in the fully ripened stage.

### 2.2. Metabolic Profiling of Fruit at Different Stages of Ripening

The polyphenolic metabolites of the fruit from three ripening stages were studied based on UPLC-MS/MS and databases. The repeatability and reliability of our method was confirmed by overlapping display total ion chromatograms (TICs) detected and analyzed by mass spectrometry of different QC (quality control) samples. The results show that the curves of TIC detected have high overlap (Appendix A), indicating that the signal stability is good when the same sample is detected by mass spectrometry at different times. Furthermore, MRM (Multiple reaction monitoring) results are shown in Appendix A. Each mass spectrum peak with different colors represents a metabolite detected. In total, 531 polyphenolic metabolites were detected, including 220 phenolic acids (41.43%) and 219 flavonoids (41.24%, 17 anthocyanidins, 10 chalcones, 22 flavanols, 19 flavanones, 11 flavanonols, 48 flavones, 83 flavonols, and 9 isoflavones), 50 tannins (9.42%, 41 tannins and 9 proanthocyanidins) and 42 lignans and coumarins (7.91%) (Table 2). Detailed information is shown in Appendix A. The retention time, declustering potential (DP) and collision energy (CE) of 30 key metabolites is provided in Appendix A.

### 2.3. PCA and OPLS-DA Analysis of Differentially Accumulated Metabolites (DAMs)

A principal component analysis (PCA) was used to observe the degree of variation between different groups. We extracted two principal components, PC1 and PC2, which were 32.17% and 16.95%, respectively, (Figure 2A). In the PCA score plot, G and R were clearly separated. However, the samples of R and P were close to each other in the direction of PC1, which indicated that the samples of G differed from those of R and P while the difference between R and P was not significant.

Orthogonal signal correction and partial least squares-discriminant analysis (OPLS-DA) can maximize the differences between groups and serves as an effective method to screen different metabolites. Q^2^ is a significant parameter to evaluate in the OPLS-DA models, which indicates the predictive ability of the model. When Q^2^ > 0.5, it can be considered to be an effective model. In our research (Figure 2B,C), the OPLS-DA model compared the polyphenol metabolic content from rosehips in different stages to evaluate the difference between G and R (R^2^X = 0.548, R^2^Y = 0.997, Q^2^ = 0.755) and between R and P (R^2^X = 0.557, R^2^Y = 0.99, Q^2^ = 0.586), which demonstrated that these models are effective at predicting, reliable and can be utilized to search for additional differential polyphenol metabolites.

### 2.4. Differential Metabolic Profiling during Fruit Growth and Development

We combined the fold change and adopted the variable importance in project (VIP) value of the OPLS–DA model to screen DAMs between different ripening stages. Fold change values ≥ 2 or ≤0.5 and VIP ≥ 1 were considered to denote significant differences.

In total, there were 160 DAMs between G and R, and there were 99 upregulated metabolites and 61 downregulated metabolites (Figure 3A). Notably, 50 and 107 metabolites of the 157 DAMs identified between R and P were upregulated and downregulated, respectively, (Figure 3B). This indicated that most DAMs were upregulated during the first stage and downregulated during the second stage of ripening. The first 20 DAMs with the largest VIP value in the OPLS-DA model are displayed in Figure 3C,D. The first 20 DAMs with the largest VIP value in the immature to semi-mature stages were all significantly upregulated. Among the top 20 DAMs with the largest VIP value in the semi-mature to mature stages, 13 metabolites were significantly downregulated, and phenolic acids (10) comprised a large part. Furthermore, we selected fifteen DAMs and showed their differences in Figure 3E.

Phenolic acids were the most abundant metabolites detected in our research. From G to R, 39 compounds were downregulated, and 29 were upregulated. From R to P, 50 compounds were downregulated, and 17 compounds were upregulated. Gallic acid did not change significantly in the first stage but decreased significantly from G to R (Figure 3E). In addition, 76 gallic acid derivatives were detected, and 31 of them were downregulated with fruit ripening. Only two compounds (1-*O*-Galloyl-2-*O*-p-Coumaroyl-β-d-glucose and gallocatechin-gallocatechin) were upregulated from immature to semi-mature stage.

Flavonols (83) were the most abundant flavonoids observed in this study, quercetin glycoside, kaempferol derivatives and isorhamnetin were more abundant. From G to R, all differential flavonols metabolites (26) were upregulated, while from R to P, 25 flavonols were downregulated (all the differential kaempferol metabolites were downregulated) and five were upregulated, including four quercetin metabolites. In this study, rutin did not change significantly during fruit ripening, while hyperin increased during the early stage of fruit ripening and stabilized during the late stage of fruit ripening (Figure 3E).

Seventeen anthocyanins were detected in this experiment, with ten upregulated from G to R and four were upregulated from R to P. Only one was downregulated from R to P. Four anthocyanin differential metabolites, including cyanidin-3-*O*-arabinoside, pelargonidin-3-O-glucoside, cyanidin-3-O-glucoside (Figure 3E) and Cyanidin-3-O-(6″-O-malonyl)glucoside, upregulated during whole ripening stage.

## 3. Discussion

In recent decades, rose hips have grown in popularity owing to their large number of biologically active compounds, such as ascorbic acid, carotenoids and phenolic compounds [17]. The interest in plant polyphenols derives from the evidence of their potent antioxidant activity, and their wide range of pharmacologic properties that include anti-inflammatory, antiallergic and antibacterial activities [9]. The total contents of polyphenols from rose hips (84.6–174.8 mg g^−1^) is higher than those of some fruit and berries, such as blueberry (*Vaccinium corymbosum* v. Jersey, 2.70–3.48 mg g^−1^), strawberry (*Rubus idaeus* v. *Tulameen,* 1.61–2.94 mg g^−1^) and raspberry (*Rubus idaeus* v. *Tulameen*, 2.70–3.03 mg g^−1^) [18,19]. In addition, rose hips have a higher total antioxidant capacity compared with other fruits, including sour cherry (*Prunus cerasus*), blackberry (*Rubus nemoralis*), strawberry (*Fragaria vesca*), and raspberry (*Ru. idaeus* ssp. *vulgatus*) [20]. Currently, the research and product development of rose hips are only concentrated on a few species, such as *R. canina*, *R. rugosa*, *R. roxburghii* and *R. laevigata*. The components of many rose hips are not clearly understood. The lack of scientific evidence of their nutritional value and health care efficacy has prohibited the promotion and application of these wild fruit in the modern food industry. Therefore, the purpose of this study was to develop extensive knowledge of the phenolic metabolites and their changes during the fruit development and the maturation of *R. xanthina* f. *spontanea* to lay a foundation for utilization in the food, nutrition and cosmetic industries.

Widely targeted metabolites based on UPLC-MS/MS have been extensively applied in the study of mechanisms of human diseases and their diagnoses, changes in the active ingredients of functional foods under different treatments, and metabolic changes in plants under different conditions or in different genotypes [21], owing to its high-throughput, high degree of sensitivity, wide coverage, and qualitative and quantitative accuracy [14]. In the previous study, cyanidin-3-glucoside (cy-glu) was the only anthocyanin identified in all the hip samples [11,22,23]. However, 17 anthocyanins were detected in this experiment. The data measured by UPLC-MS/MS technology are more extensive, which helps us to understand the pattern of phenolic profile in *R. xanthina* f. *spontanea* fruit and comprehensively analyze the changes of compounds during fruit ripening, which provides a basis for the production of rose hips.

The phenolic compound of a plant depends on the plant genotype and environmental factors during growth and postharvest treatment [24]. In our study, overall, more metabolites between unripe and half-ripe rose hips were upregulated and more metabolites between half-ripe and fully ripe rose hips were downregulated. Only seven flavonoids were downregulated in first stage, compared with 67 upregulated. After that, 40 out of 68 differential flavonoids were downregulated. By contrast, more phenolic acids were downregulated throughout the whole period.

Phenolic acids are considered to be one of the functional food groups of components in fruit and are thought to contribute to the health effects of plant-derived products by eliminating free radical species, inhibiting free radical formation, and preventing oxidative damage to DNA [25]. Phenolic compounds are the most abundant secondary metabolites in the *R. xanthina* f. *spontanea* fruit detected in this study. Rose hips are rich in a variety of phenolic acids, and chlorogenic acid, gallic acid, caftaric acid, p-coumaric acid, and ferulic acid were found to be the principal components in rose hips [2,26], which were also found in our experiments. Among them, gallic acid and its derivatives and chlorogenic acid were relatively high in green hips (Figure 3E). Gallic acid has many biological characteristics, including antioxidant, anti-inflammatory and antibacterial properties, and chlorogenic acid has similar effects [27,28]. Moreover, recent study found that gallic acid could exert anti-cancer activities via several biological pathways, including migration, metastasis, apoptosis, cell cycle arrest, angiogenesis, and oncogene expression [27].

Rose hips have been reported to be rich in catechin, rutin, quercetin and kaempferol [2,17]. Flavonols were the most abundant flavonoids observed in this study. No flavonols were significantly downregulated from immature to semi-mature fruit. In this study, quercetin and kaempferol were the primary components. Several biological effects of quercetin and kaempferol have been demonstrated in vitro and in vivo, which include antioxidation, anti-inflammatory, anticancer, and antidiabetic activities [29,30]. No anthocyanins were significantly downregulated during maturity except petunidin-3-*O*-(6″-*O*-caffeoyl) glucoside, which was significantly downregulated from R to P. In addition, the changes in color of the *R. xanthina* f. *spontanea* fruit during ripening could be related to the accumulation of anthocyanins.

## 4. Materials and Methods

### 4.1. Plant Materials

The fruit of *R. xanthina* f. *spontanea* were collected from the beginning of June to the end of July 2021 from bushes growing on the campus of Beijing Forestry University, Beijing, China (116.35° E, 40.01° N). The rose hips were collected at three ripening stages: (1) unripe (green and hard, G), according to BBCH scale stage 79,709 (fruits have reached 90% of final size) (2) half-ripe (red and hard, R), according to BBCH scale stage 85,805 (increasing intensity of fruit color), and (3) fully ripe (purple and soft, P), according to BBCH scale stage 88,808 (full ripeness: cultivar-/species-specific fruit coloring and seed ripeness) [31]. Fruit (20 hips per plant) from each ripening stage were randomly collected from three plants that were propagated by asexual reproduction. After they were collected, the seeds and calyx were removed. After being frozen immediately in liquid nitrogen, they were transported to the laboratory and stored at −80 °C until analysis. The experimental research on plants, including the collection of plant materials, conforms to relevant institutional, national, and international guidelines and legislation.

### 4.2. Rose Hip Color Measurements

The color of rose hips was measured using a portable colorimeter (NF555, Nihon Dempa Kogyo Co., Ltd., Tokyo, Japan), which had been adjusted with a white standard calibration plate before use. We used the CIELab system to describe colors, where the color parameter *L** ranged from 0 (black) to 100 (white), corresponding to a dark-bright scale and representing the relative lightness. The color parameters *a** and *b** range from −60 to 60, where *a** is negative for green; *a** is positive for red, and *b** is negative for blue and positive for yellow. The hue angle (*h*) is expressed in degrees from 0 to 360. Thirty rose hips were measured per stage.

### 4.3. Sample Preparation and Extraction

The samples were prepared as described by Xu et al. [32] with modifications. The freeze-dried fruit were ground into powder using a Mixer Mill for 1.5 min at 30 Hz (MM 400; Retsch, Haan, Germany), and approximately 100 mg of fruit powder was extracted with 1.2 mL of 70% aqueous methanol overnight at 4 °C. After centrifugation at 12,000 rpm for 10 min, the extracts were filtered using a 0.22 μm pore size filter (SCAA-104; ANPEL, Shanghai, China) before further analysis.

### 4.4. Conditions for Chromatography-Mass Spectrometry

The analytical conditions and related parameters were conducted as previously described by Li et al. [33]. The extracts were analyzed using an UPLC-MS/MS system (UPLC, Nexera X2; Shimadzu, Tokyo, Japan; MS, 4500 Q TRAP; Applied Biosystems, Waltham, MA, USA). The analytical conditions are illustrated in Table 3: column, Agilent SB-C18 (1.8 µm, 2.1 mm × 100 mm); The mobile phase was instituted of solvent A, pure water with 0.1% formic acid, and solvent B, acetonitrile with 0.1% formic acid. Sample measurements were performed with a gradient program that employed the starting conditions of 95% A, 5% B. Within 9 min, a linear gradient to 5% A, 95% B was programmed, and a composition of 5% A, 95% B was kept for 1 min. Subsequently, a composition of 95% A, 5.0% B was adjusted within 1.1 min and kept for 2.9 min. The flow velocity was set as 0.35 mL per minute; The column oven was set to 40 °C; The injection volume was 4 μL. The effluent was alternately connected to an ESI-triple quadrupole-linear ion trap (QTRAP)-MS. The MS analysis was performed with electrospray ionization (ESI) at 500 °C and 5500 V (positive ion mode)/−4500 V (negative ion mode); ion source gas I (GSI), gas II (GSII), and curtain gas (CUR) were set at 50, 60, and 25.0 psi, respectively; the collision-activated dissociation (CAD) was high. Instrument tuning and mass calibration were performed with 10 and 100 μmol/L polypropylene glycol solutions in QQQ and LIT modes, respectively. QQQ scans were acquired as MRM experiments with collision gas (nitrogen) set to medium. The declustering potential (DP) and collision energy (CE) for individual MRM transitions were conducted with further DP and CE optimization. A specific set of MRM transitions were monitored for each period based on the metabolites eluted within this period.

### 4.5. Qualitative and Quantitative Analysis of Metabolites

The metabolites were qualitatively and quantitatively analyzed as described by Li et al. [33] and Wang et al. [34]. We used high-resolution mass spectrometry AB sciex 6600 QTOF for qualitative detection of mixed samples (Appendix A), and then use AB sciex4500 QTRAP for relative quantification of samples. Based on the self-built database MWDB (MetWare Biological Science and Technology Co., Ltd., Wuhan, China), the compounds were qualitatively analyzed by comparing the accurate precursor ions (Q1), production (Q3) values, and retention time (RT).

In MRM mode, the quadrupole screened for the parent ions of target substances and eliminated any ions derived from substances of different molecular weights to preliminarily eliminate interference. Further, the precursor ions were fragmented to form many fragment ions. The characteristic ions of each metabolite were screened through the QQQ mass spectrometer to obtain the signal intensity and eliminates the interference of non-target ions, so that the quantification is more accurate and the repeatability is better. After obtaining the mass spectrometry data of metabolites of different samples, the peak area integration of mass spectrometry peaks of all substances was carried out. The chromatographic peaks were integrated and corrected by MultiaQuant (AB SCIEX; Framingham, MA, USA), and the relative contents of corresponding compounds were represented as chromatographic peak area integrals.

### 4.6. Statistical Analysis

Three biological replicates were analyzed for each experiment. PCA and OPLS-DA were performed using R (https://www.r-project.org) (accessed on 18 September 2021) to predict the stability and reliability of the model. Multi-dimensional statistical analyses (using the VIP value and fold change) were used to select DAMs. Statistical analyses were performed using Microsoft Office Excel 2016 (Redmond, WA, USA) and SPSS 23.0 (IBM, Inc., Armonk, NY, USA). The pictures were drawn by GraphPad Prism 9.0 (GraphPad Software, San Diego, CA, USA).

## 5. Conclusions

A total of 531 polyphenolics were identified by UPLC-MS/MS. Phenolic acids (41.4%), and flavonoids (41.2%) were the primary components. Gallic derivatives, quercetin glycoside, kaempferol derivatives and anthocyanins were rich in content, which have been shown to have a significant impact on human health. Flavonoids are upregulated more from the immature to semi-mature stages and downregulated more from the semi-mature to mature stages. Phenolic acids are significantly downregulated in both stages, and there is a greater variety of them. These data comprehensively analyzed the changes of polyphenolics during the development of rose hips for the first time, which lays a foundation for the commercial production and basic research on the metabolism of rose hips.

## Figures and Tables

**Figure 1 metabolites-12-00438-f001:**
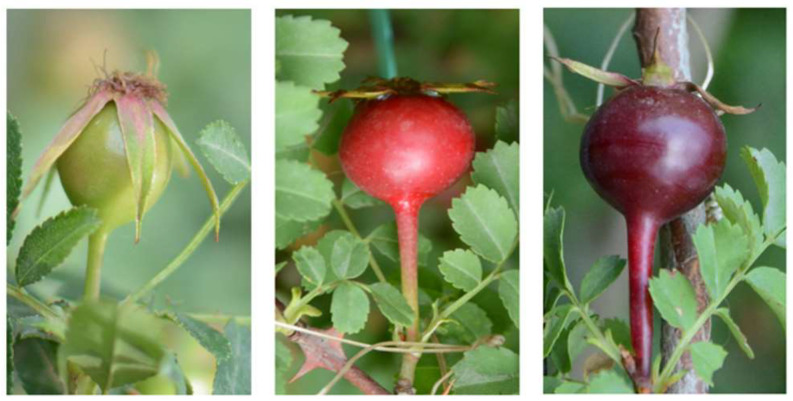
Images of three ripening stages of *Rosa xanthina* f. *spontanea*.

**Figure 2 metabolites-12-00438-f002:**
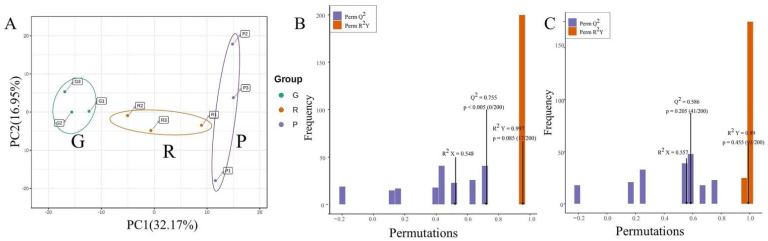
(**A**) PCA score plot of the metabolites in different ripening stages of rose hips. (**B**) OPLS-DA analyses of Green hips and Red hips. (**C**) OPLS-DA analyses of Red hips and Purple hips. Note: G: green hips, unripe; R: red hips, half ripe; P: purple hips, fully ripe.

**Figure 3 metabolites-12-00438-f003:**
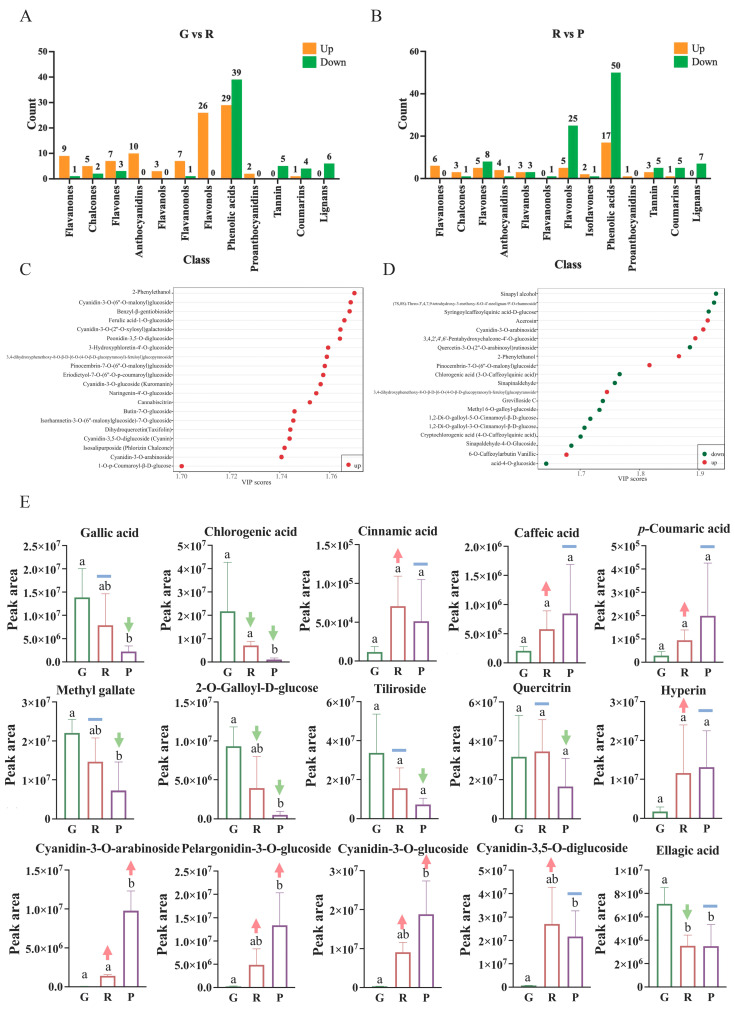
(**A**) The counts of differential metabolites in G vs. R. (**B**) The counts of differential metabolites in R vs. P. (**C**) Top 20 VIP metabolites in G vs. R. (**D**) Top 20 VIP metabolites in R vs. P. Ordinate: metabolite, Blackish green color: downregulated metabolites, Red: upregulated metabolites. (**E**) Changes in the major polyphenols of fruit at different developmental stages (Different lowercase letters indicate significant differences (*p* value < 0.05); the red arrow indicates that the substance is up-regulated at this stage, the green arrow indicates that the substance is down-regulated at this stage (Fold change values ≥ 2 or ≤0.5 and VIP ≥ 1), and the blue horizontal line indicates that it is not up-regulated or down-regulated.). Note: G: green hips, unripe; R: red hips, half ripe; P: purple hips, fully ripe.

**Table 1 metabolites-12-00438-t001:** Fruit color parameters of *Rosa xanthina* f. *spontanea* hips during ripening.

	*L**	*a**	*b**	*C*	*h*
Green hips	63.60 ± 2.6 c	−8.81 ± 2.8 a	28.51 ± 3.7 c	29.92 ± 4.10 b	106.95 ± 4.34 a
Red hips	39.82 ± 5.21 b	26.40 ± 6.91 c	9.25 ± 6.08 b	28.67 ± 6.75 b	19.21 ± 11.99 b
Purple hips	31.13 ± 3.9 a	6.16 ± 2.0 b	1.43 ± 3.1 a	6.77 ± 2.78 a	10.44 ± 17.30 b

Note: Mean ± SD; Different lowercase letters indicate significant differences. *p* < 0.05.

**Table 2 metabolites-12-00438-t002:** Classification of polyphenol metabolites in *Rosa xanthina* f. *spontanea* hips.

Class I	Class II	Number of Metabolites
Flavonoids	Flavanones	19
	Chalcones	10
	Flavones	48
	Anthocyanidins	17
	Flavanols	22
	Flavanonols	11
	Flavonols	83
	Isoflavones	9
Phenolic acids	Phenolic acids	220
Tannins	Proanthocyanidins	9
	Tannins	41
Lignans and Coumarins	Coumarins	17
	Lignans	25
Total		531

**Table 3 metabolites-12-00438-t003:** Analytical conditions of the Ultra Performance Liquid Chromatography (UPLC).

Conditions	Parameters	
Column	Agilent SB-C18 (1.8 µm, 2.1 mm × 100 mm)	
Mobile phase	Mobile phase A (pure water with 0.1% formic acid) Mobile phase B (acetonitrile with 0.1% formic acid)	
Gradient program	0 min	95:5 *v/v* (Mobile phase A: Mobile phase B)
	0–9 min	a linear gradient to 5:95 *v*/*v*
	9–10 min	5:95 *v*/*v*
	10–11.1 min	adjust to 95:5 *v*/*v*
	11.1–14 min	95:5 *v*/*v*
Flow rate	0.35 mL/min	
Column temperature	40 °C	
Injection volume	4 μL	

## Data Availability

The data presented in this study are available in Appendix A.

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
