# Peer review of "Widely Targeted Metabolic Profiling Reveals Differences in Polyphenolic Metabolites during Rosa xanthina f. spontanea Fruit Development and Ripening"

_metabolites, 2022, doi:10.3390/metabo12050438_

Round 1

Reviewer 1 Report

Comments for authors
This paper entitled “Widely Targeted Metabolic Profiling Reveals Differences in Polyphenolic Metabolites during Rosa xanthina f. spontanea Fruit Development and Ripening” by Sun et al. reported the work in the targeted metabolomic analysis of 531 polyphenol metabolites from Rosa xanthina f. spontanea fruits. In addition, the authors investigated the changes in polyphenolic metabolism during fruit development. However, several questions should be addressed. 
1.    The authors claimed this metabolomic study will help select the optimal harvest time of rose hips to achieve better quality. What are the current criteria to identify the maturity of rose hips? Is there any question in current methods to identify the maturity of rose hips? The authors should explain why this work is helpful for the selection of the optimal harvest time of rose hips.
2.    The authors mentioned the MRM transition list in Lines 274-277. However, this list provided in supporting information is not completed. The information on retention time, declustering potential (DP), and collision energy (CE) for individual MRM transitions are missing. 
3.    In Lines 282-283, the authors described “The metabolites were qualitatively and quantitatively analyzed as described by Li et al.[34] and Wang et al.[35].”, so, whether was the high-resolution MS used in the identification of the targeted metabolites? Or is the QQQ-MS/MS used in the qualitative analysis of polyphenolic metabolites? The authors should specify them clearly to avoid confusion. In Lines 286-288, the authors claimed that “Isotope signals, repetitive signals containing K+, Na+, NH4+, and repetitive signals of fragment ions of other large molecular weight were removed during identification.”, thus, the authors should specify how to remove the interferences which were from the isotopic ions, adduct ions, and in-source fragmentation products by using QQQ-MS/MS.
4.    In Lines 294-295, did the authors do HCA and the enrichment analysis of metabolic differences? The authors should show the results and discuss them.
5.    In Figure 3E, is the peak area used as the relative content of metabolite? The bar chart showed a huge variation in the quantitative results of intra-group samples. Could the authors provide the method validation data to support that the variation of metabolites levels of intra-group samples is from the divergence of rose hips itself? If yes, I do think that only three biological replicates are insufficient for the comparative quantitative analysis of polyphenol metabolites of rose hips during fruit development.
6.    Lines 70-78 should be deleted.
7.    There are two Table 2! In the first Table 2, “All” should be changed to “Number of metabolites”.
8.    In Figure 2, “G”, “R”, and “P” should be specified clearly in the figure legend.
9.    Lines 14 and 66, “ultra-performance liquid chromatography tandem mass spectrometry” may be changed as ultra-performance liquid chromatography-tandem mass spectrometry”.
10.    In the whole manuscript, the abbreviations should be spelled out when they appear in the main text for the first time.
In conclusion, the manuscript represents some work of potential significance but still has many open questions and an unclear identification of metabolites.

Author Response

Reviewer 1

Thank you very much for your kind comments about our manuscript. These comments are exceedingly valuable and very helpful for revising and improving our manuscript. In the subsequent text, we seek to answer your questions and provide a detailed account of the changes made to the original manuscript.

  1.    The authors claimed this metabolomic study will help select the optimal harvest time of rose hips to achieve better quality. What are the current criteria to identify the maturity of rose hips? Is there any question in current methods to identify the maturity of rose hips? The authors should explain why this work is helpful for the selection of the optimal harvest time of rose hips.

Thank you for your significant reminder. There is a criteria of growth stages of roses and we have added this reference in section 4.1 plant materials (Line 240-244). There isn’t any problem about the criteria. However, the polyphenolic changes in unripe, half-ripe, and ripe rose hips has not been evaluated. Such information could add value to rose hips, which could increase their commercialization and consumption.

  1.    The authors mentioned the MRM transition list in Lines 274-277. However, this list provided in supporting information is not completed. The information on retention time, declustering potential (DP), and collision energy (CE) for individual MRM transitions are missing. 

We have added the graph of MRM results as supplementary figure 2 in section 2.2(Line 97-99). Please check this graph in our supplementary file.

We have the information on DP and CE. If necessary, we can submit it as supplementary table.

  1.    In Lines 282-283, the authors described “The metabolites were qualitatively and quantitatively analyzed as described by Li et al.[34] and Wang et al.[35].”, so, whether was the high-resolution MS used in the identification of the targeted metabolites? Or is the QQQ-MS/MS used in the qualitative analysis of polyphenolic metabolites? The authors should specify them clearly to avoid confusion.

Thank you for your significant reminder.

We use high-resolution mass spectrometry AB sciex 6600 QTOF for qualitative detection of mixed samples, and then use AB sciex4500 QTRAP for relative quantification of samples,(Line294-296) combining the advantages of non-targeted and targeted metabolomics, using high-resolution Mass spectrometry for accurate characterization, with triple quadrupole mass spectrometry as a complementary tool with high sensitivity, high specificity, and excellent quantitative capabilities.

In Lines 286-288, the authors claimed that “Isotope signals, repetitive signals containing K+, Na+, NH4+, and repetitive signals of fragment ions of other large molecular weight were removed during identification.”, thus, the authors should specify how to remove the interferences which were from the isotopic ions, adduct ions, and in-source fragmentation products by using QQQ-MS/MS.

As follows, we added more details to section 4.3 Qualitative and Quantitative Analysis of Metabolites. (Line 302-309)

In MRM mode, the quadrupole screened for the parent ions of target substances while eliminates any ions derived from substances of different molecular weights to preliminarily eliminate interference. Further, the precursor ions were fragmented to form many fragment ions. The characteristic ions of each metabolite were screened through the QQQ mass spectrometer to obtain the signal intensity and eliminates the interference of non-target ions, so that the quantification is more accurate and the repeatability is better. After obtaining the mass spectrometry data of metabolites of different samples, the peak area integration of mass spectrometry peaks of all substances was carried out.

  1.    In Lines 294-295, did the authors do HCA and the enrichment analysis of metabolic differences? The authors should show the results and discuss them.

Thank you for pointing it out. It is a mistake to mention HCA here. We are not going to mention this analysis in the results, so we have deleted HCA here.

  1.   In Figure 3E, is the peak area used as the relative content of metabolite? The bar chart showed a huge variation in the quantitative results of intra-group samples. Could the authors provide the method validation data to support that the variation of metabolites levels of intra-group samples is from the divergence of rose hips itself? If yes, I do think that only three biological replicates are insufficient for the comparative quantitative analysis of polyphenol metabolites of rose hips during fruit development.

Yes, peak area was used as the relative content of metabolite. Actually, we mentioned this in part 4.3 ‘The chromatographic peaks were integrated and corrected by MultiaQuant (AB SCIEX; Concord, Canada), and the relative contents of corresponding compounds were represented as chromatographic peak area integrals.’ (Line 309-312)

We selected the materials from three samples of cuttage propagation, and the morphological characteristics of the fruits are similar, however, bar chart showed a huge variation in the quantitative results of intra-group samples. Therefore, in the next research, we will collect more individuals and biological replicates for further research.

  1.    Lines 70-78 should be deleted.
    7.    There are two Table 2! In the first Table 2, “All” should be changed to “Number of metabolites”.
    8.    In Figure 2, “G”, “R”, and “P” should be specified clearly in the figure legend.
    9.    Lines 14 and 66, “ultra-performance liquid chromatography tandem mass spectrometry” may be changed as ultra-performance liquid chromatography-tandem mass spectrometry”.

Thank you for your reminding, we have corrected these mistakes.

  1.    In the whole manuscript, the abbreviations should be spelled out when they appear in the main text for the first time.

Thanks a lot.

In 2.1 Measurement of color, we have added ‘The rose hips were collected at three ripening stages: 1) unripe (green and hard, G), 2) half-ripe (red and hard, R), and 3) fully ripe (purple and soft, P).’ (Line 74-76)

In 2.3, we removed DAMs, and replaced it with ‘differentially accumulated metabolites’ (Line 105).

Reviewer 2 Report

The manuscript reports an ultra-performance liquid chromatography tandem mass spectrometry (UPLC-MS/MS) method for a widely targeted metabolomics analysis on the polyphenolic components of Rosa xanthina f. spontanea in three ripening stages, namely unripe, half-ripe and fully ripe fruit. In total, 531 polyphenol metabolites were detected, there were 160 differential metabolites between unripe and half-ripe rose hips and 157 differential metabolites between half-ripe and fully ripe rose hips.

The work is interesting and the results achieved in this study can provide metabolomics information on the changes in polyphenolic metabolism during fruit development. However, there some important aspects that need to be clarified prior to further evaluation.

- Please delete lines 70-78.

2.2 Metabolic Profiling of fruit at different stages of ripening. Please report a chromatogram highlighting at least the most abundant peaks. Despite a classification of polyphenol metabolites in Rosa xanthina f. spontanea hips are reported in Table 2, the authors need to report the identity of all compounds detected in a separated Table (As supplementary material). How was identification performed?

2.4 Differential Metabolic Profiling during Fruit Growth and Development. How was quantification carried out? Concerning quantification, as reported in Section 3.3, the metabolites were quantified using the MRM mode. This aspect needs to be critically described also because no reference material is reported in section 4. Materials and Methods.

3.2 Conditions for Chromatography-Mass Spectrometry. Please report full description for the UPLC-MS/MS system. Further, specify the specific set of MRM transitions.

Please pay attention in numbering of subsections. There are wrong numbers in Section 4. 4. Materials and Methods.

Author Response

Thank you very much for your kind comments about our manuscript. These comments are exceedingly valuable and very helpful for revising and improving our manuscript. In the subsequent text, we seek to answer your questions and provide a detailed account of the changes made to the original manuscript.

- Please delete lines 70-78.

Thanks a lot, we have deleted it.

2.2 Metabolic Profiling of fruit at different stages of ripening. Please report a chromatogram highlighting at least the most abundant peaks.

Thank you for your reminder. We have added the graph of total ion chromatograms (TICs) and MRM results in supplementary figure in section 2.2.(Line 92-99)

Despite a classification of polyphenol metabolites in Rosa xanthina f. spontanea hips are reported in Table 2, the authors need to report the identity of all compounds detected in a separated Table (As supplementary material). How was identification performed?

We have presented a supplementary table 1, showing details of all compounds.

The identification method was described in Line 296-301. Further, MRM (Multiple reaction monitoring) results are shown in Supplementary Figure 2. Each mass spectrum peak with different colors represents a metabolite detected.

 2.4 Differential Metabolic Profiling during Fruit Growth and Development. How was quantification carried out? Concerning quantification, as reported in Section 3.3, the metabolites were quantified using the MRM mode. This aspect needs to be critically described also because no reference material is reported in section 4. Materials and Methods.

Thank you for your significant reminder. As follows, we have added more details to section 4.3 Qualitative and Quantitative Analysis of Metabolites as follows. (Line 302-312)

In MRM mode, the quadrupole screened for the parent ions of target substances while eliminates any ions derived from substances of different molecular weights to preliminarily eliminate interference. Further, the precursor ions were fragmented to form many fragment ions. The characteristic ions of each metabolite were screened through the QQQ mass spectrometer to obtain the signal intensity and eliminates the interference of non-target ions, so that the quantification is more accurate and the repeatability is better. After obtaining the mass spectrometry data of metabolites of different samples, the peak area integration of mass spectrometry peaks of all substances was carried out. The chromatographic peaks were integrated and corrected by MultiaQuant (AB SCIEX; Concord, Canada), and the relative contents of corresponding compounds were represented as chromatographic peak area integrals.

3.2 Conditions for Chromatography-Mass Spectrometry. Please report full description for the UPLC-MS/MS system. Further, specify the specific set of MRM transitions.

As follows, we added more details to section 4.2.(Line271-278)

The analytical conditions are illustrated in Table 3: column, Agilent SB-C18 (1.8 µm, 2.1 mm * 100 mm); The mobile phase was instituted of solvent A, pure water with 0.1% formic acid, and solvent B, acetonitrile with 0.1% formic acid. Sample measurements were performed with a gradient program that employed the starting conditions of 95% A, 5% B. Within 9 min, a linear gradient to 5% A, 95% B was programmed, and a composition of 5% A, 95% B was kept for 1 min. Subsequently, a composition of 95% A, 5.0% B was adjusted within 1.1 min and kept for 2.9 min. The flow velocity was set as 0.35 mL per minute; The column oven was set to 40°C; The injection volume was 4 μL.

What’s more, We have added the graph of MRM results in supplementary figure in section 2.2.(Line 92-99).

Please pay attention in numbering of subsections. There are wrong numbers in Section 4. 4. Materials and Methods.

The mistake has been corrected, thank you.

Round 2

Reviewer 1 Report

1. The authors should provide the completed MRM transition list including the information of retention time, declustering potential (DP), and collision energy (CE) for individual metabolites. Because it is very critical for the quantitative data acquisition by using LC-QQQ-MS/MS. (supplementary figure 2 is insufficient).

2. The authors used high-resolution mass spectrometry AB sciex 6600 QTOF for qualitative detection of mixed samples in this work (Line294-296). Thus, the authors should added this experiment description (such as the QTOF MS conditions) in Materials and Methods or Supplementary file.

3. In my opinion, it does not make sense to use quadrupole to preliminarily eliminate interference of isotopic ions and in-source fragmentation products without corresponding standards (the unique retention time, the accurate mass obtained from high-resolution MS). In addition, if there is insufficient chromatographic separation, the characteristic ions of metabolite and interference of in-source fragmentation products, which obtained from the product ion scan, were almost the same. Thus, I do not suggest the authors claim that “Isotope signals, repetitive signals containing K+, Na+, NH4+, and repetitive signals of fragment ions of other large molecular weight were removed during identification.”, or the authors should explain them clearly because the explanation in Lines 302-309 is insufficient (only QQQ-MS/MS is insufficient).

4. According to "Yes, peak area was used as the relative content of metabolite. Actually, we mentioned this in part 4.3 ‘The chromatographic peaks were integrated and corrected by MultiaQuant (AB SCIEX; Concord, Canada), and the relative contents of corresponding compounds were represented as chromatographic peak area integrals.’ (Line 309-312)", I would like to suggest the authors use "Peak area" instead of "Relative content" in Figure 3E.

5. Based on the authors' response, i.e., "We selected the materials from three samples of cuttage propagation, and the morphological characteristics of the fruits are similar, however, bar chart showed a huge variation in the quantitative results of intra-group samples. Therefore, in the next research, we will collect more individuals and biological replicates for further research.", did the authors have QC samples data to track or evaluate the quality of the quantitative results of the targeted metabolites? Or can the authors illustrate the reliability of the quantification of the targeted metabolites? Because it is critical for the conclusion in this work. 

Author Response

  1. The authors should provide the completed MRM transition list including the information of retention time, declustering potential (DP), and collision energy (CE) for individual metabolites. Because it is very critical for the quantitative data acquisition by using LC-QQQ-MS/MS. (supplementary figure 2 is insufficient).

We have presented a supplementary table 1(Line 103), showing details of all compounds. The retention time as well as DP and CE of thirty key metabolites is provided in Supplementary table 2(Line 105).

  1. The authors used high-resolution mass spectrometry AB sciex 6600 QTOF for qualitative detection of mixed samples in this work (Line294-296). Thus, the authors should added this experiment description (such as the QTOF MS conditions) in Materials and Methods or Supplementary file.

We have added a supplementary table 3,4 (Line 294) to describe identification method. Please check it.

  1. In my opinion, it does not make sense to use quadrupole to preliminarily eliminate interference of isotopic ions and in-source fragmentation products without corresponding standards (the unique retention time, the accurate mass obtained from high-resolution MS). In addition, if there is insufficient chromatographic separation, the characteristic ions of metabolite and interference of in-source fragmentation products, which obtained from the product ion scan, were almost the same. Thus, I do notsuggest the authors claim that “Isotope signals, repetitive signals containing K+, Na+, NH4+, and repetitive signals of fragment ions of other large molecular weight were removed during identification.”, or the authors should explain them clearly because the explanation in Lines 302-309 is insufficient (only QQQ-MS/MS is insufficient).

We have deleted it, thanks.

  1. According to "Yes, peak area was used as the relative content of metabolite. Actually, we mentioned this in part 4.3 ‘The chromatographic peaks were integrated and corrected by MultiaQuant (AB SCIEX; Concord, Canada), and the relative contents of corresponding compounds were represented as chromatographic peak area integrals.’ (Line 309-312)", I would like to suggest the authors use "Peak area" instead of "Relative content" in Figure 3E.

Thanks a lot. We have changed the y-axis from "Relative content" to "Peak area" in Figure 3E.

  1. Based on the authors' response, i.e., "We selected the materials from three samples of cuttage propagation, and the morphological characteristics of the fruits are similar, however, bar chart showed a huge variation in the quantitative results of intra-group samples. Therefore, in the next research, we will collect more individuals and biological replicates for further research.", did the authors have QC samples data to track or evaluate the quality of the quantitative results of the targeted metabolites? Or can the authors illustrate the reliability of the quantification of the targeted metabolites? Because it is critical for the conclusion in this work. 

Thank you for your reminder. We have added the graph of total ion chromatograms (TICs) in supplementary figure in section 2.2.(Line 92-97).

The repeatability and reliability of our method was confirmed by overlapping display total ion chromatograms (TICs) detected and analyzed by mass spectrometry of different QC (quality control) samples. The results show that the curves of TIC detected have high overlap (Supplementary Figure 1), indicating that the signal stability is good when the same sample is detected by mass spectrometry at different times.

Reviewer 2 Report

The authors have adequately addressed all Reviewers remarks and the improved paper can be now accepted in the present form.

Author Response

Thank you.

Round 3

Reviewer 1 Report

The authors have satisfactorily addressed most of my concerns. In particular, the authors have added the critical method information for LC-MS data acquisition of the key metabolites in the revised manuscript, however, it will be perfect if the authors can provide the basic acquisition information for all metabolites identified in this work.